# Ocular Surface Changes Differ Significantly Between Oxaliplatin- and Diabetes-Induced Polyneuropathy

**DOI:** 10.3390/ijms26051884

**Published:** 2025-02-22

**Authors:** Martin Schicht, Marco Sisignano, Jessica Farger, Saskia Wedel, Nichapa Phunchago, Natarajan Perumal, Caroline Manicam, Klaus Scholich, Gerd Geisslinger, Friedrich Paulsen, Elke Lütjen-Drecoll

**Affiliations:** 1Institute of Functional and Clinical Anatomy, Friedrich-Alexander University Erlangen-Nürnberg, 91054 Erlangen, Germanynichph@kku.ac.th (N.P.); friedrich.paulsen@fau.de (F.P.); elke@drecoll.net (E.L.-D.); 2Institute of Clinical Pharmacology, Goethe University, 60590 Frankfurt am Main, Germany; marco.sisignano@med.uni-frankfurt.de (M.S.); scholich@em.uni-frankfurt.de (K.S.); gerd.geisslinger@itmp.fraunhofer.de (G.G.); 3Fraunhofer Institute for Translational Medicine and Pharmacology (ITMP), 60590 Frankfurt am Main, Germany; 4Department of Anatomy, Khon Kaen University, Khon Kaen 40002, Thailand; 5Department of Ophthalmology, University Medical Centre of the Johannes Gutenberg University Mainz, 55131 Mainz, Germany; nperumal@eye-research.org (N.P.); caroline.manicam@unimedizin-mainz.de (C.M.)

**Keywords:** oxaliplatin (Ox), polyneuropathies (PNs), ocular surface, dry eye disease (DED), tear film

## Abstract

Dry eye disease (DED) is often seen in patients with polyneuropathies (PNs), but the relationship between the different forms of PNs and DED is not known. In oxaliplatin (Ox-)-treated mice with PNs, morphological changes in the sciatic nerve (SN), dorsal root ganglia (DRG), trigeminal ganglia (TG), and the ocular tissues involved in tear formation were investigated. In addition, the tear proteomics and the gene expression of related proteins in the ocular surface tissues as well as inflammatory factors were analyzed. There were significant changes in six tear proteins compared to the controls, with respective changes in gene expression in the ocular tissues. Morphologically, there was a decrease in the number of conjunctival goblet cells and changes in the myofibroblasts surrounding the Meibomian glands. The lacrimal gland appeared normal. In the SN, there was a slight decrease in the number of mitochondria without signs of inflammation. In the DRG, 30–50% of the small- and medium-sized neuronal cells had swollen mitochondria. In contrast, the mitochondria of the TG were unremarkable. The changes in the tear film proteins and the ocular tissue morphology involved in tear formation in OPN differed significantly from those previously described in DPN mice, despite a similar mechanical hypersensitivity and similar morphological features of the DRG. In DPN, these changes led to aqueous-deficient dry eye disease, whereas in OPN, they resulted in evaporative DED. Furthermore, in contrast to DPN, the TG in OPN showed no morphological alterations, which indicates differences in the peripheral nerve changes and ocular nerve damage between the two conditions.

## 1. Introduction

The platinum-based drug oxaliplatin is used as a first-line chemotherapy treatment for colorectal cancer in the FOLFOX scheme [1]. Depending on the dose and duration of treatment, acute or chronic forms of oxaliplatin-induced polyneuropathy (OPN) can develop [2]. On the other hand, only a proportion of chemotherapy-treated patients develop persistent OPN, and to date, predicting the risk remains challenging, even within groups who receive the same drug regimen. Patients with OPN suffer a loss of sensory function and sensory pain beginning in the lower extremities. Also, cold allodynia, the painful response to a normally nonpainful cold stimulus, is a characteristic feature of OPN [3]. Changes in the autonomic nervous system have also been observed. In addition, DED is a common complication of polyneuropathies (PNs) such as OPN. However, the underlying pathogenesis of OPN is unknown. Confocal microscopy of the corneal nerves in patients with OPN has been performed, but the results have been heterogeneous [4,5,6]. Furthermore, the current protocol for determining the optimal oxaliplatin dose for individual patients involves determining the severity of chronic peripheral neuropathy based only on symptoms. In 2015, a large clinical trial with oxaliplatin-treated patients was conducted and correlative evidence between early oxaliplatin-induced acute pain and the occurrence of chronic OPN was reported. The authors observed that patients with a strong initial response to oxaliplatin and severe oxaliplatin-induced acute pain were more likely to develop a severe, chronic neuropathy after treatment [7]. However, objective measurements of peripheral nerve structure and function, including skin biopsies, do not appear to provide clinically relevant evidence of peripheral nerve degeneration [8]. Therefore, it remains to be investigated if there are markers that can be used for the early diagnosis and risk assessment for the development of chronic OPN.

To better understand the pathomechanisms of OPN and to find objective markers for the development of OPN during treatment, we investigated peripheral nerve fibers, DRGs, and ocular surface changes in mice with oxaliplatin-chemotherapy-induced OPN in this study. We analyzed the number of mitochondria in peripheral nerve fibers, the morphology of DRG neurons, and the concentrations of chemokines, cytokines, and growth factors in nervous tissue. We also investigated the changes in the tear film proteome and the tissue morphology involved in tear film production, including the conjunctiva, meibomian, and lacrimal glands, and compared these changes with those of the sciatic nerve and the spinal and trigeminal ganglia. As early as 8 to 10 days after the onset of treatment, there were signs of OPN in the peripheral nerves and concomitant ocular surface changes, including changes in tear proteins. An investigation of tear proteins and ocular surface changes in oxaliplatin-treated patients might provide a new tool for the early diagnosis of OPN during chemotherapy.

Our findings also showed that the changes in the tear film proteins as well as the morphological changes in the ocular surface tissues of oxaliplatin-treated mice differed from the changes that have been previously reported in mice with streptozotozin-induced diabetic PNs (STZ-mice, DPN model) [9]. Therefore, the non-invasive investigation of tear proteins and conjunctival goblet cells and confocal microscopy of the corneal nerves can help to develop a better understanding of the pathogenesis of DED in combination with PNs.

## 2. Results

### 2.1. Mechanical Pain Threshold

Initially, the impact of oxaliplatin treatment (3 mg/kg oxaliplatin i.p.) on the mechanical pain thresholds of male wild-type mice (C57BL/6N) was evaluated on the treatment day and again ten days later using a dynamic plantar aesthesiometer. The mice treated with oxaliplatin exhibited significantly shorter paw withdrawal latencies on day 10 (Figure 1).

### 2.2. Quantitative Changes in the Morphology of the Sciatic Nerve

The axons and Schwann cells of all non-myelinated and myelinated nerves had the same appearance in the treated mice and the controls (Figure 2A). Although the neurofilaments and the mitochondria looked normal, there was a significant decrease in the number of mitochondria in the myelinated nerves. In the non-myelinated nerves, there was no significant decrease in the number of mitochondria (Figure 2B). The axon diameters did not change compared to the controls after 5 or 10 days of treatment (Figure 2C).

### 2.3. Endoneurium and Perineural Sheath

Within the endoneurium, the endothelium of the capillaries was stained for endostatin and showed intact junctional complex staining for ZO1. There was no edema and no increase in free cells within the endoneurium. The connective tissue surrounding the axons was unchanged (Figure 3A).

The perineuronal sheath in the treated mice had the same appearance as in the controls. The different layers of cells that were stained for alpha smooth muscle actin were connected by tight junctions (zonulae occludentes) staining for ZO1, and they were surrounded by basement membranes (Figure 3B).

### 2.4. Dorsal Root Ganglia (DRG)

The number and distribution of small neuronal cells staining for isolectin B4, medium-sized cells staining for calcitonin gene-related peptide (CGRP), and large DRG staining for neurofilament H were unchanged between the treated and control mice. However, within the cytoplasm of several cells, the mitochondria were significantly swollen, while the other organelles looked unchanged. Cells with swollen mitochondria were located next to cells with normal mitochondria, indicating that fixation artefacts were not causative of the changes (Figure 4A). The number of damaged cells increased with the duration of treatment. After 5 days, 31.99% of the small and 56.8 ± 15% of the medium-sized-cells were affected. The appearance of all the large cells was normal. After 10 days, 42.3 ± 21% of the small, 67.6 ± 1% of the medium-sized, and 10% of the large cells showed swollen mitochondria (Figure 4B). Damaged cells were seen in the center and at the rim of the DRG.

### 2.5. Concentrations of Proinflammatory Cytokines, Chemokines, and Growth Factors

The concentrations of proinflammatory cytokines, growth factors, and chemokines were analyzed in the DRG, sciatic nerve, and dorsal spinal cord tissue from oxaliplatin- or vehicle-treated mice (3 mg kg^−1^) ten days after treatment. No differences could be detected in the levels of fibroblast growth factor beta (FGF-β), tumor necrosis factor alpha (TNF-α), vascular endothelial growth factor (VEGF), or the interleukins IL-2, IL-4, IL-5, IL-6, or IL-17 when comparing oxaliplatin- and vehicle-treated mice (Figure 5).

Likewise, concentrations of the chemokines CC-chemokine ligands 2 and 3 (CCL2 and CCL3) were analyzed in the nervous tissue of oxaliplatin- and vehicle-treated mice and did not reveal any differences between the treatment groups (Figure 6).

However, differences in the expression of different neuronal and oxidative stress markers of oxaliplatin- and vehicle-treated animals were observed in the dorsal root ganglia and spinal cord. We assessed the expression of the neuronal stress markers activating transcription factor 3 and matrix metallopeptidase 9, as well as the oxidative stress markers NADPH oxidase 2 (NOX 2), NADPH oxidase 4 (NOX 4), and xanthine dehydrogenase, ten days after treatment (Figure 7). Based on this, we observed a significant decrease in both xanthine dehydrogenase (Figure 7B) and NOX 2 in the spinal cord of oxaliplatin-treated animals compared to vehicle-treated animals. In contrast, we observed an increase in the expression of NOX 4 in the dorsal root ganglia.

### 2.6. Ocular Surface Changes

#### 2.6.1. Tear Film Proteome

After 10 days of treatment, a total of 141 tear proteins with a false discovery rate of <1% were identified in both the OPN and their controls. Among these, seven proteins were found to be significantly differentially abundant in the OPN tears compared to the control (Figure 8A). A significant increase in the abundance of biorientation of chromosomes in cell division protein 1-like 1 (BOD1L1; *p* = 9.9 × 10^−4^), keratin, type II cytoskeletal 1 (KRT1; *p* = 8.9 × 10^−3^), protein C19orf12 homolog (1600014C10Rik; *p* = 2.4 × 10^−2^), lactotransferrin (Ltf; *p* = 4.8 × 10^−2^), and lactoperoxidase (Lpo; *p* = 1.2 × 10^−2^) was observed in the OPN group. On the contrary, the probable cation-transporting ATPase 13A2 (ATP13A2; *p* = 2.3 × 10^−2^) and homeobox EHOX (Rhox4b; *p* = 4.6 × 10^−2^) proteins were found to be decreased in abundance in the OPN tears compared to the control (Figure 8B,C). Since there is no human ortholog for Rhox4b and 1600014C10RIK, they were not included in further analyses.

#### 2.6.2. Immunohistochemistry

The five proteins BOD1L1, ATP13A2, Lpo, Ltf, and KRT1 were localized in different tissues involved in the production of tear film. ATP13A2 and Lpo reactivity was seen in the cornea, conjunctiva, meibomian glands, and lacrimal gland; Ltf staining was observed in the conjunctiva; KRT1 staining was observed in the cornea, conjunctiva, and lacrimal gland; and BODL1L1 staining was observed in the corneal endothelium and conjunctiva (as an example, Figure 9 shows staining in the conjunctival epithelium).

#### 2.6.3. Gene Expression

A gene expression analysis of Lpo, Ltf, ATP13A2, BOD1L1, and KRT1 was performed in the related ocular tissues of OPN animals compared to controls. For Lpo, a significant decrease was seen in the cornea (* *p* < 0.05), a significant increase was seen in the lacrimal gland (** *p* < 0.01), and no change was observed in the eyelid. For Ltf, there was a significant decrease in the cornea (** *p* < 0.01). ATP13A2 showed a significant increase in the lacrimal gland (** *p* < 0.01), while the corneal expression was unchanged. The KRT1 and BOD1L1 expression in the ocular surface was significantly increased in the eyelids (* *p* < 0.05), while all other tissues revealed no obvious changes (Figure 10A–E).

### 2.7. Evaluation of Morphological Changes in the Tissues Involved in Tear Film Formation—Qualitative and Quantitative

#### 2.7.1. Conjunctival Goblet Cells

The number of goblet cells was significantly reduced in the treated mice (Figure 11A–C). Furthermore, within the goblet cells, the apical secretory vesicles were fused (Figure 11D).

#### 2.7.2. Lacrimal Gland

There were no morphological differences between treated mice and their controls. There was no increase in inflammatory cells, and the secretory endpiece cells and the duct cells appeared normal (Figure 12A). The area covered basally by rER compared to the area covered by apical vesicles showed no significant differences relative to the controls (Figure 12B). The measurement was carried out on sections from three animals of each group (with and without treatment).

#### 2.7.3. Meibomian Gland

The meibomian gland cells of Ox-treated mice looked normal, but there were changes in the surrounding connective tissue (CT) and the basement membrane (BM) of the gland (Figure 13). The BM was thickened in some places. In the CT of treated mice, there were elongated smooth muscle cells that followed the course of the basal cells of the gland (Figure 13A). These cells showed an increase in rER and a densification of the myofilaments, which is a sign of a stress reaction (Figure 13B).

#### 2.7.4. Cornea

There was no significant decrease in the concentration of tubulin in the cornea. The larger nerves (myelinated and non-myelinated) entering the cornea appeared ultrastructurally unchanged compared to the controls. Corneal wholemounts stained for beta-III tubulin revealed no changes in the nerves (Figure 14). Unfortunately, we were not able to obtain results for the concentrations of corneal CGRP.

#### 2.7.5. Trigeminal Ganglion

Serial semithin sections through the trigeminal ganglia showed no differences between the Ox-treated mice and the controls. Immunohistochemically, there were no differences in staining for CGRP and tubulin (Figure 15A). Ultrastructurally, none of the differently sized neuronal cells showed mitochondrial swellings (Figure 15B–C).

## 3. Discussion

Our findings show, for the first time, that ocular surface changes differ significantly in mouse models with OPN compared to DPN, whereas the early changes seen in the lower limbs were less different. The behavioral changes in the paw withdrawal experiments were comparable. Similar to DPN, oxaliplatin treatment caused a decrease in the mechanical pain threshold 10 days after treatment. In addition, our results show for the first time that the mitochondrial changes in the dorsal root ganglia of the mice in the two PN models were very similar. In both DPN and OPN, approximately 30–50% of the small (<300 μm^2^) isolectin B4-positive and medium-sized (300–700 μm^2^) CGRP-positive neuronal cells showed marked mitochondrial swelling in the first two weeks after treatment. In contrast, large (>700 μm^2^) NFH-positive cell bodies did not exhibit signs of swelling. The distribution of damaged neurons within the ganglia was also similar in OPN and DPN, indicating that both noxious agents reach the cells via the same route. The pathophysiology of these changes is still not clarified, but it is well established that treatment with oxaliplatin leads to mitochondrial damage [10,11,12]. Therefore, similar to STZ-induced PNs, oxidative stress is discussed as being causative for mitochondrial changes. In line with this, we observed an increase in the expression of NADPH oxidase 4 (NOX4), a major source for reactive oxygen species, in the dorsal root ganglia of oxaliplatin-treated mice. NOX4 has previously been connected with neuropathic pain after a peripheral nerve injury [13,14]. However, unlike in DPN, the sciatic nerve in OPN revealed no morphological signs of inflammation and the inflammatory chemokines, cytokines, and growth factors were not altered in the nerve or in the associated neurons when compared to the controls. This suggests that OPN is not primarily driven by inflammatory processes. Instead, it aligns with previous findings indicating a significant neuronal contribution, such as aberrant ion channel activity, mitochondrial dysfunction, or DNA damage, to OPN [15,16]. The number of mitochondria in the sciatic nerve was significantly reduced in the OPN animals. Since the structure of the mitochondria showed no differences to the controls, the reduction was probably not due to the degeneration of the cell organelles, but potentially due to impaired axoplasmic transport leading to decreased energy availability.

In contrast to the lower limbs and their innervation, changes in the ocular surface and the structures involved in tear production were significantly different in the two PN models. In Ox-treated mice, compared to the controls, the alterations in tear film protein expression differed from that previously reported in STZ mice. Specifically, our findings in the Ox-treated mice suggest a significant increase in the abundance of BOD1L1, KRT1, Ltf, Lpo, and ATP13A2, indicating degenerative changes in the corneal epithelial cells. In contrast to the STZ mice, inflammatory protein expression was absent in the tears of Ox-treated mice.

In mice with DPN, as in patients with DPN, we found a decrease in corneal nerves. It is well described that the disturbance of corneal nerves induces changes in lacrimal gland secretion [17,18,19,20], which were seen in our DPN model. In OPN with no obvious changes in the corneal nerves in the early stage of the disease, the lacrimal gland appeared normal. To investigate the pathophysiology of these differences in corneal innervation, we compared the morphological changes of the trigeminal ganglia. In DPN, they showed similar mitochondrial swelling in neuronal cells to that in the dorsal root ganglia. In OPN, the ultrastructure of the mitochondria in the trigeminal ganglia was not different compared to the controls, and immunohistochemically, there were no staining differences for CGRP. The mechanisms underlying the preserved morphology of trigeminal ganglia in early-stage OPN remain unclear. A significant decrease in conjunctival goblet cells was only seen in OPN. Further investigations are required to determine whether this is due to direct damage caused by the Ox treatment, changes in autonomic nerves, or both.

The different changes in the ocular surface structures in OPN and DPN induce different forms of DED. DED is caused by changes in the three major components of the tear film: 1. The basal layer: composed of mucins (glycoproteins), mainly secreted by conjunctival goblet cells, but also by the lacrimal and accessory lacrimal glands as well as in shedded form coming from the corneal and conjunctival epithelial cells. 2. The aqueous layer: produced by the lacrimal glands and accessory lacrimal glands (glands of Krause and Wolfring). 3. The lipid layer (most superficial component): secreted as meibum by the holocrine meibomian glands. The International Dry Eye Workshop classifies DED into two subgroups, comprising the aqueous deficiency dry eye disease (ADDED) and evaporative dry eye disease (EDED) [21]. Our findings reveal that Ox treatment with reduced goblet cells in the conjunctiva and changes in the meibomian glands (presumably caused by an obstruction to secretion, but without changes in the lacrimal glands) induces the development of EDED. In contrast, in DPN with changes in the secretory cells of the lacrimal glands and signs of reduced tear secretion (aqueous component), but without changes in the conjunctival goblet cells or the meibomian gland, ADDED is induced instead of EDED [22,23]

Clinically, the volume of tear fluid, the determination of proteins in the tear fluid, and the amount of goblet cells in the conjunctiva and the corneal nerves can be determined non-invasively. These findings can facilitate the specific, early diagnosis of PNs in clinical practice and guide targeted treatment strategies for DED.

## 4. Materials and Methods

All the animal experiments were approved by the local ethics committees for animal research (Darmstadt, Germany) under the approval code FK/1113 and FU/2018. In addition, all the animal experiments were performed according to the Working Group PPRECISE (Preclinical Pain Research Consortium for Investigating Safety and Efficacy) [24] and the recommendations of the Guide of the Care and Use of Laboratory Animals of the National Institute of Health and the ARRIVE guidelines [25]. All the experimental C57BL/6NRj animals were purchased from the commercial breeding company Janvier (Le Genest-Saint-Isle, France). They were housed in a day/night cycle of a 12 h rhythm and food and water were available ad libitum. In addition, all the animal experiments were performed with healthy, male, 8- to 12-week-old C57BL/6NRj mice.

### 4.1. Oxaliplatin-Induced Neuropathic Pain Model

First, an oxaliplatin (Cayman Chemical, Ann Arbor, MI, USA) stock solution of 3 mg/mL in autoclaved deionized water was prepared. Then, the oxaliplatin stock solution of 3 mg/mL was diluted to 1:4 in saline (sodium chloride 0.9% (*v*/*v*); Fresenius Kabi, Bad Homburg, Germany). Finally, a dose of 3 mg kg^−1^ of the 1:4 diluted oxaliplatin stock solution or vehicle (saline; 0.9% sodium chloride) was injected intraperitoneally (i.p.) in 8- to 12-week-old male C57BL/6NRj animals, as described previously [26]. Thus, the application of the described oxaliplatin-induced neuropathic pain model was in accordance with the suggestions of the PPRECISE Working Group [24]. Only mice with OPN were investigated. For the experiments, we used healthy adult C57/Bl6N wild-type mice that were tumor-free before the oxaliplatin treatment. They did not receive any treatment other than oxaliplatin, which was administered solely to induce oxaliplatin-induced neuropathy, following the protocol described by Nassini et al., 2011 [26]. The health status of the animals was monitored daily following the oxaliplatin treatment. No changes in behavior, signs of bleeding, inflammation, or discomfort were observed. Additionally, the animals were weighed daily, and none of the treated mice exhibited a decrease in body weight after the oxaliplatin treatment. Eight animals per group (control vs. OPN) were used for this study.

### 4.2. Behavioral Experiments

Before starting the behavioral experiments and treatment phases, the locomotor function of all the animals was verified using a rotarod assay [27]. During all the behavioral experiments, the experimenter was blinded. For assessing the mechanical pain threshold, the animals were placed at least one hour before the measurement in the respective test cages to allow habituation. After this habituation phase, the mechanical paw withdrawal threshold was assessed using a dynamic plantar aesthesiometer (Hargreaves apparatus, Ugo Basile, Varese, Italy). The time until paw withdrawal was determined by applying linear ascending force onto the plantar surface at the same rate for each trial (0–5 g over 10 s) with a cutoff time of 20 s, as described previously [28]. Baseline measurements were performed on two consecutive days before the treatment phase.

### 4.3. Multiplex Immunoassay

For performing the ProcartaPlex assay (Thermo Fisher Scientific, Rockford, IL, USA), the dorsal root ganglia (DRGs), the sciatic nerve, and lumbar segments of the spinal cord (SC) were collected from vehicle- or oxaliplatin-treated mice ten days after treatment. Tissue samples were collected and homogenized and the assay was performed as previously described [29]. Sciatic nerve samples were resuspended in 100 µL, DRG samples in 120 µL, and spinal cord samples in 300 µL of assay lysis buffer. The tissue was homogenized as described and the samples were measured according to the manufacturer’s instructions.

### 4.4. Murine Tear Sample Preparation and Protein Extraction

The label-free quantification of peptides via one-dimensional electrophoresis (1DE) and a liquid chromatography (LC)–electrospray ionization (ESI)-MS/MS strategy was employed to identify the changes in the proteome in the designated groups [30,31,32,33,34]. The tear samples (23 µg per sample with three biological replicates) were subjected to 1DE using 4–12% Bis-Tris Gels (Invitrogen, Karlsruhe, Germany) with the MEPS running buffer under reducing conditions for 60 min with a constant voltage of 150 V, according to the manufacturer’s instructions. The SeeBlue Plus 2 (Invitrogen, Karlsruhe, Germany) pre-stained protein standard was used as a molecular mass marker. Next, the gels were stained with a Colloidal Blue Staining Kit (Invitrogen, Karlsruhe, Germany) according to the manufacturer’s instructions. The visualized protein bands were sliced into 8 gel pieces for each group and were cut into small pieces and subjected to dehydration by utilizing neat acetonitrile prior to disulfide bond cleavage with 10 mM 1,4-dithiothreitol (DTT) in 100 mM ammonium bicarbonate (NH_4_HCO_3_) and alkylation with 55 mM iodoacetamide (IAA) in 100 mM NH_4_HCO_3_. The reduced and alkylated protein mixtures were digested with sequence-grade trypsin for 16 h at 37 °C. Subsequently, proteolysis was quenched by the acidification of the reaction mixtures with 200 μL of extraction buffer composed of 1:2 (*vol*/*vol*) 5% formic acid/acetonitrile, and the mixtures were incubated for 15 min at 37 °C in a shaker. The supernatant containing peptides was collected and the remaining peptides in the gel pieces were extracted with two 20 min washes in extraction buffer. The supernatants were pooled and concentrated to dryness in a SpeedVac (Eppendorf, Darmstadt, Germany) prior to storage at −20 °C [31,33,35]. Next, the peptides recovered from the in-gel digestion were subjected to purification employing SOLAµ™ SPE-Plate (Thermo Fisher Scientific, Rockford, IL, USA) according to the manufacturer’s instructions. This peptide-purification procedure was repeated three times for each sample, and the combined eluate was dried in the SpeedVac and dissolved in 10 μL of a 0.1% trifluoroacetic acid (TFA) solution prior to the LC-MS/MS analysis.

### 4.5. Liquid Chromatography–Electrospray Ionization–MS/MS (LC-ESI-MS/MS) Analysis

The LC-ESI-LTQ-Orbitrap MS system is well established in our laboratory and details of this system are described in detail elsewhere [36,37,38,39]. Briefly, solvent A, which consisted of LC-MS-grade water with 0.1% (*v*/*v*) formic acid, and solvent B, consisting of LC-MS-grade acetonitrile with 0.1% (*v*/*v*) formic acid, were utilized. The gradient was run for 60 min per gel band. Continuum mass spectra data were acquired on an ESI-LTQ-Orbitrap-XL MS (Thermo Scientific, Bremen, Germany) The LTQ-Orbitrap was operated in a data-dependent mode of acquisition to automatically switch between Orbitrap-MS and LTQ-MS/MS acquisition. Full-scan MS survey spectra (from *m*/*z* 300 to 2000) were acquired in the Orbitrap with a resolution of 30,000 at *m*/*z* 400 and a target automatic gain control (AGC) setting of 1.0 × 10^6^ ions. The five most intense precursor ions were sequentially isolated for fragmentation in the LTQ with collision-induced dissociation (CID) fragmentation; the normalized collision energy (NCE) was set to 35% with an activation time of 30 ms, a repeat count of 3, and a dynamic exclusion duration of 600 s. The resulting fragment ions were recorded in the LTQ.

### 4.6. Label-Free Quantification (LFQ) and Statistical Analysis

The acquired continuum MS spectra were analyzed by the MaxQuant computational proteomics platform, version 1.6.3.3, and its built-in Andromeda search engine for peptide and protein identification [40,41,42,43]. The tandem MS spectra were searched against several databases as follows: Swiss-Prot, homo sapiens, date—17 October 2018, annotated proteins—20410; Swiss-Prot, mus musculus, date—17 October 2018, annotated proteins—17001; Swiss-Prot, rattus, date—17 October 2018, annotated proteins—8050; TrEMBL, homo sapiens, date—30 November 2018, annotated proteins—156062; TrEMBL, mus musculus, date—2 May 2018, annotated proteins—67764. Standard settings with a peptide mass tolerance of ±20 ppm, a fragment mass tolerance of ±0.5 Da, ≥6 amino acid residues, and only “unique plus razor peptides” that belong to a protein were chosen. For limiting a certain number of peak matches by chance, a target-decoy-based FDR for peptide and protein identification was set to 0.01. The carbamidomethylation of cysteine was set as a fixed modification, while protein N-terminal acetylation and the oxidation of methionine were defined as variable modifications; enzyme: trypsin and maximum number of missed cleavages: The output of the generated “proteingroups.txt” data from the MaxQuant analysis was utilized for a subsequent statistical analysis with the Perseus software (version 1.6.1.3) [44]. In the Perseus software, the statistical analysis was performed with the following parameters: First, a log_2_ transformation of all the protein intensities was performed. For statistical evaluation, Student’s two-sided *t*-test was used for all the groups’ comparisons, with *p* ˂ 0.05 used to identify the significantly differentially abundant proteins.

### 4.7. Morphology

For transmission electron microscope (TEM) and light microscope investigations, the animals were perfusion-fixed with 4.5% paraformaldehyde (PFA), and the dorsal root ganglia (L1–L6), the distal portion of the sciatic nerve (distal from the branching into the tibial and fibular nerves), the proximal portion (next to the spinal cord), the eyes with eyelids, the lacrimal glands, and the trigeminal ganglia were removed. For immunohistochemistry, specimens of the different tissues were deep-frozen or embedded in paraffin. For TEM investigations, all the tissues were postfixed in ITO’s fixative [45] and embedded in Epon in the usual way. Semithin (1 µm) and ultrathin sections were cut with a microtome (Ultracut Leica, Jena, Germany). The semithin sections were stained with toluidine blue. The ultrathin sections were analyzed with a JEM-1400Plus (JEOL, Tokyo, Japan).

For immunofluorescence, sections were incubated with the antibodies listed in Table 1 (for a detailed description of the method, see Schicht et al., 2024 [9]).

### 4.8. Quantitative Evaluation of Goblet Cells

Serial sections of 10um thick were cut through the middle portion of the eyelids of 3 Ox-treated mice and 3 controls (around 230 sections/eye). The sections were stained with PAS (periodic acid-Schiff reaction) and the goblet cells were counted with the Leica DME light microscope (Leica Microsystems, Switzerland). This procedure is described in detail by Welss et al., 2021 [46].

### 4.9. Quantitative Analysis of Semithin and Ultrathin Sections

A quantitative analysis of the surface, the diameter and the counting was performed by using the software SightX-Viewer (Jeol, Version 1.2.3.537) or open source ImageJ (version 1.50i, National Institute of Health (NIH), Bethesda, MD, USA).

### 4.10. Gene Expression Analysis

The sciatic nerve, dorsal root ganglia, spinal cord, and ocular tissues (cornea, eyelids, and extra orbital lacrimal glands) were isolated and stored at −20 °C. This method has been described in detail by Schicht et al., 2024 [9]. The cDNA from OPN and control mice were used to perform a quantitative expression analysis. The measurements and analysis were performed in triplicate by Roche Light Cycler 480 II (Roche, Basel, Switzerland). All the primers used are listed in Table 2. 18S was carried as the housekeeping gene in each measurement. In the results, the data related to the expression of 18S are shown.

### 4.11. TaqMan^®^ Gene Expression Assay System

For assessing the gene expression of neuronal and oxidative stress markers via the TaqMan^®^ Gene Expression Assay System, the total RNA of the sciatic nerve, DRG, and SC samples were isolated using the mirVana miRNA Isolation Kit (Applied Biosystems, Thermo Fisher Scientific, Rockford, IL, USA) according to the manufacturer’s instructions. This method has been described in detail by Schicht et al., 2024 [9]. The quantitative real-time PCR was conducted with the QuantStudio™ Design & Analysis Software v 1.4.3 (Thermo Fisher Scientific) in a TaqMan^®^ Gene Expression Assay System (Table 3, Thermo Fisher Scientific) according to the manufacturer’s instructions. The raw data were evaluated using the ΔΔC(T) method, as described previously [47,48].

### 4.12. Statistical Analysis

The data are presented as the mean ± SEM. Normality was assessed using the Shapiro–Wilk test. For in vitro experiments comparing two groups, an unpaired heteroskedastic Student’s *t*-test with Welch’s correction was used. When comparing more than two groups, a one-way ANOVA was performed, while a two-way ANOVA was used for comparisons involving more than three groups. Behavioral experiments (Figure 1) were analyzed using a two-way ANOVA, followed by Bonferroni’s post hoc correction for multiple comparisons. For the proteomic analysis, Student’s two-sided *t*-test was used (Figure 8). All the statistical analyses were conducted using GraphPad Prism 9. A *p*-value < 0.05 was considered statistically significant (*p* < 0.05 = *, <0.01 = **, <0.001 = ***, n.s. = not significant).

## 5. Conclusions

Ocular surface changes are detectable after only 5–10 days of oxaliplatin treatment, which is concomitant with the occurrence of neuropathy. The ocular surface changes differ significantly in OPN and DPN, and in contrast to DPN, the trigeminal ganglia at the early onset of OPN showed no morphological changes, indicating that the pathogenesis of the ocular changes differs between the two groups. Tear fluid can easily be obtained non-invasively from patients and combined with a proteomic analysis of the tear film. Additionally, reductions in goblet cell density and corneal nerve function can also be examined through non-invasive methods. These findings may significantly improve the ability to diagnose PNs early and accurately in clinical practice, enabling more effective and targeted treatment strategies for DED.

## Figures and Tables

**Figure 1 ijms-26-01884-f001:**
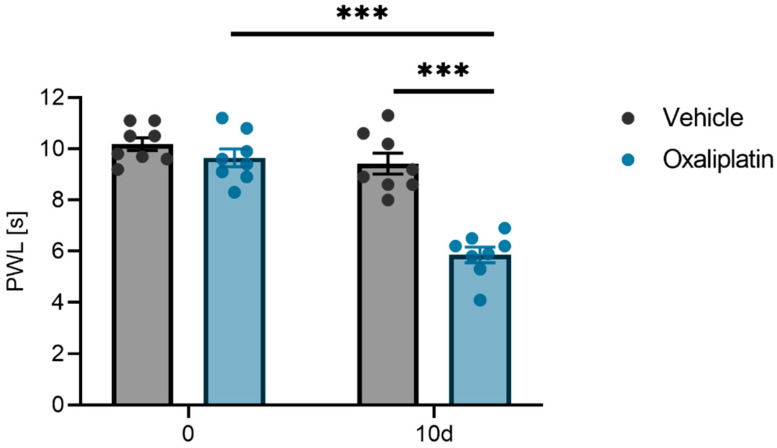
Mechanical pain thresholds of wild-type BL/6 mice before and 10 days after an i.p. injection of oxaliplatin (3 mg/kg) compared to a vehicle (0.9% *v*/*v* DMSO) using a dynamic plantar test. Shown is the mean ± SEM of the paw withdrawal latency (PWL) of n = 8 mice per group: *** *p* < 0.001 (two-way ANOVA with Bonferroni post-hoc test).

**Figure 2 ijms-26-01884-f002:**
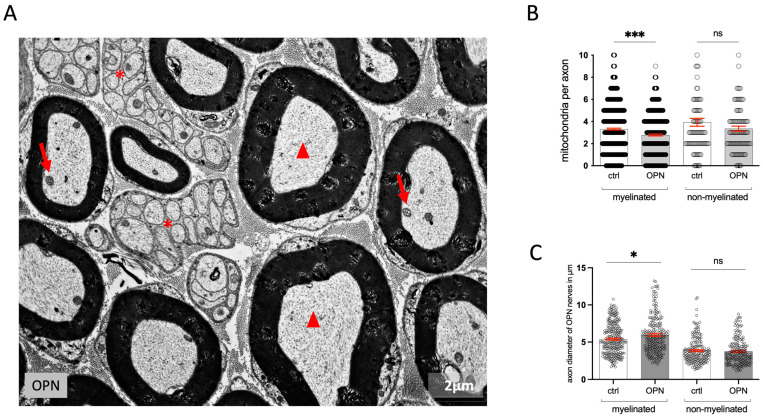
(**A**) Electron micrograph showing myelinated (arrowheads) and nonmyelinated (asterisks) nerves of an OPN mouse with numerous mitochondria in the axons (arrows). All the axons looked similar to their controls. (**B**) Axon diameters of myelinated and nonmyelinated nerves in OPN mice compared to controls, ctrl (n = 6, OPN n = 3). (**C**) Number of mitochondria per axon. The statistic is the mean ± SEM, and the points represent the mean values of individual animals: ns = not significant; * *p* < 0.05; *** *p* < 0.001 (*t*-test).

**Figure 3 ijms-26-01884-f003:**
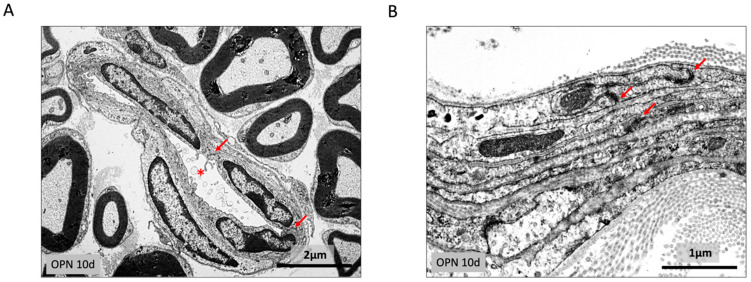
Transmission electron micrographs of (**A**) a capillary in the endoneurium (asterisk) with intact junctional complexes between endothelial cells (arrows); (**B**) the different layers of normal-looking perineuronal cells connected by zonulae occludentes (arrows).

**Figure 4 ijms-26-01884-f004:**
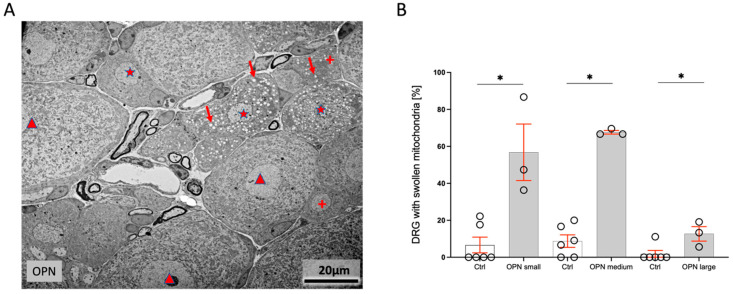
(**A**) Transmission electron micrograph of small (crosses), medium-sized (asterisks), and large (arrowheads) DRG neuronal cells in an OPN mouse. The large neuronal cells looked normal, and a proportion of the small and medium-sized mitochondria had swollen mitochondria (arrows). (**B**) Quantification of the percentage of small, medium-sized, and large neuronal cells with swollen mitochondria (ctrl, n = 6, OPN n = 3). The statistic is the mean + SEM of small (<300 µm^2^), medium (300–700 µm^2^), and large (>700 µm^2^) perikarya [%]; * *p* < 0.05 (*t*-test).

**Figure 5 ijms-26-01884-f005:**
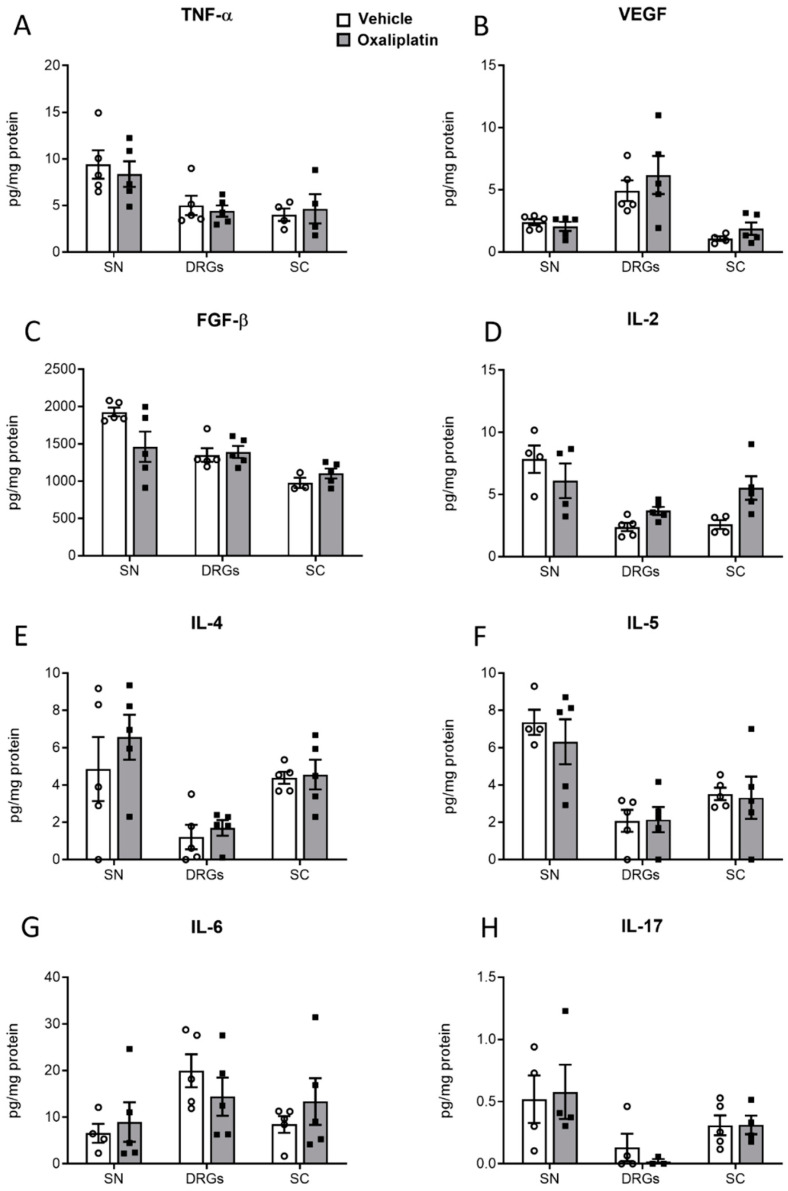
Concentrations of inflammatory growth factors and cytokines in nervous tissue of vehicle-treated (white bars) or oxaliplatin-treated (grey bars, 3 mg kg^−1^) mice. Tissue was collected ten days after the treatment. Shown are the concentrations of TNF-α (**A**), VEGF (**B**), FGF-β (**C**), IL-2 (**D**), IL-4 (**E**), IL-5 (**F**), IL-6 (**G**), and IL-17 (**H**) in the sciatic nerve (SN), dorsal root ganglia (DRG), and dorsal spinal cord (SC). Not detected were IL-1a, IL-1b, IL10, and IL-12. Shown are the results from n = 5 mice per group, measured in technical duplicate. The raw data were normalized to the total protein amount of the sample.

**Figure 6 ijms-26-01884-f006:**
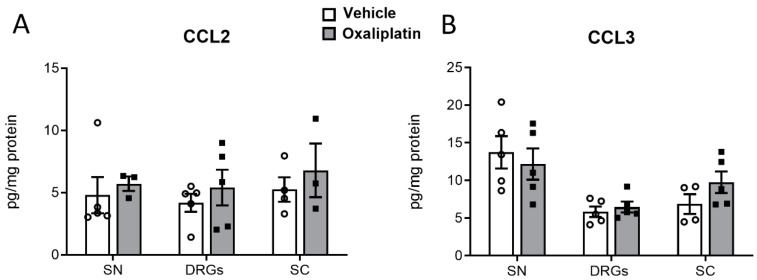
Concentrations of inflammatory chemokines in the nervous tissue of vehicle-treated (white bars) and oxaliplatin-treated (grey bars, 3 mg kg^−1^) mice. Tissue was collected ten days after the treatment. Shown are the concentrations of CCL2 (**A**) and CCL3 (**B**) in the sciatic nerve (SN), dorsal root ganglia (DRG), and dorsal spinal cord (SC). Not detected were C-X-C chemokine ligands 3 and 9 (CXCL3 and CXCL9). Shown are the results from n = 5 mice per group, measured in technical duplicate. The raw data were normalized to the total protein amount of the sample.

**Figure 7 ijms-26-01884-f007:**
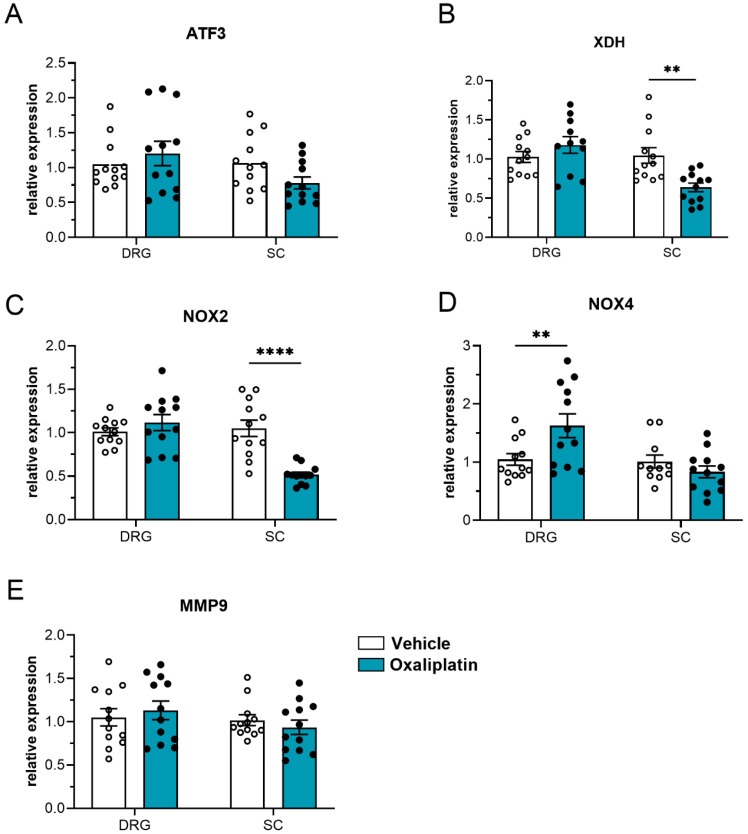
Relative expression of neuronal and oxidative stress markers ATF3 (**A**), XDH (**B**), NOX2 (**C**), NOX4 (**D**), and MMP9 (**E**) in dorsal root ganglia and spinal cord tissue of vehicle-treated (white bars) and oxaliplatin-treated (blue bars, 3 mg kg^−1^) mice. The data display the mean ± SEM from 4 mice per group, measured in technical triplet. ** *p* < 0.01, **** *p* < 0.0001 (two-way ANOVA with Sidak’s post hoc correction).

**Figure 8 ijms-26-01884-f008:**
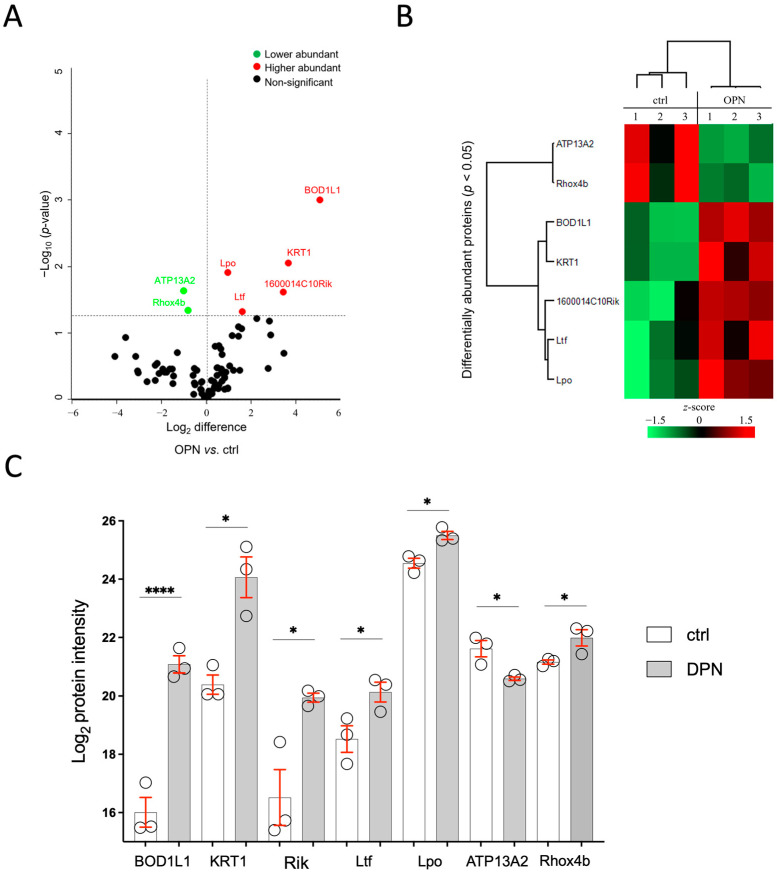
Proteomic analysis of tear film of oxaliplatin-treated mice (OPN, 8–10 d) compared with control animals (ctrl, 8–10 d), measured by LC-MS/MS; n(OPN/ctrl) = 3 animals/group. (**A**) Volcano plot and (**B**) heat map illustrating the seven significantly differentially abundant tear proteins in OPN compared to the ctrl. (**C**) Bar graph of log_2_ intensity of OPN and the ctrl. Bars represent the mean ± SEM, * *p* < 0.05, **** *p* < 0.001 (Student’s two-sided *t*-test).

**Figure 9 ijms-26-01884-f009:**
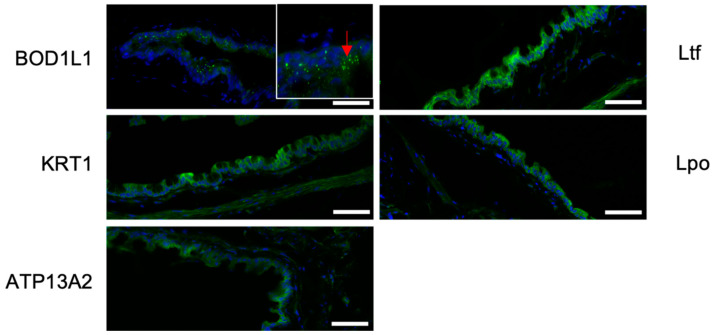
Immunohistochemical detection (green) of Bod1l (red arrow), Atp13a2, Lpo, LTF, and Krt1 in the conjunctiva of wild-type mice. Nuclei are visible with DAPI (blue); scale bar: 50 µm.

**Figure 10 ijms-26-01884-f010:**
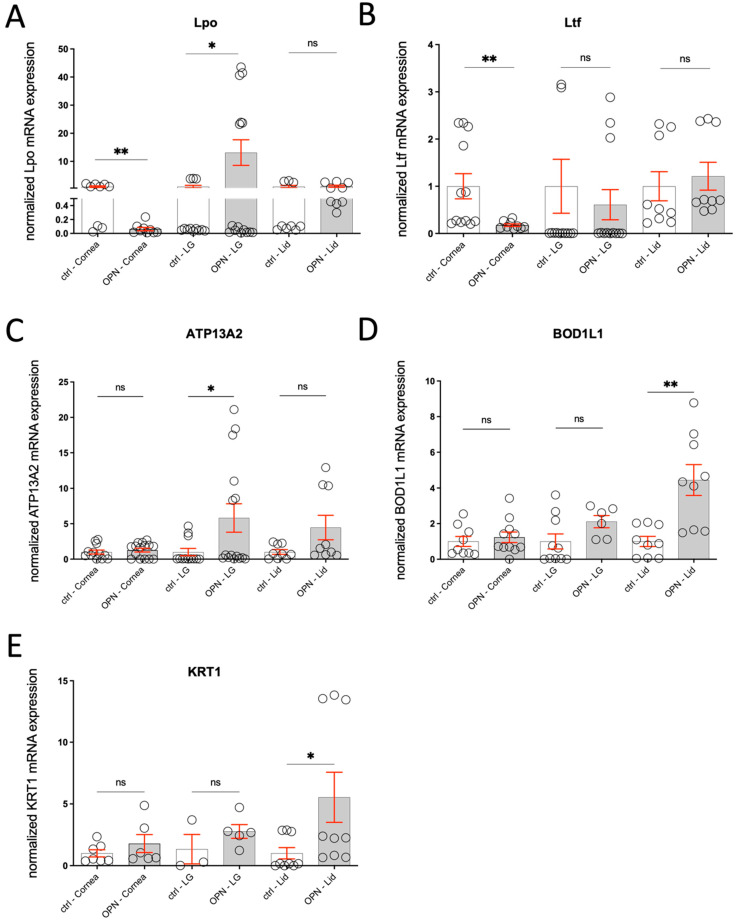
Diagrams showing the differences in gene expression in OPN animals (n = 3–7) of (**A**) Lpo, (**B**) Ltf, (**C**) ATP13A2, (**D**) BOD1L1, and (**E**) KRT1 in the cornea, lacrimal gland, and eye lid. The statistic is the mean + SEM relative to the mean of the control (n = 3–5); * *p* < 0.05, ** *p* < 0.01, ns = not significant (*t*-test).

**Figure 11 ijms-26-01884-f011:**
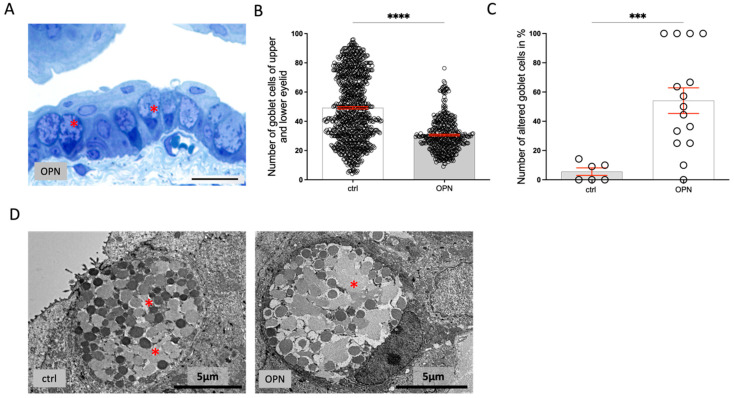
(**A**) A 1 μm section through the conjunctiva with altered goblet cells (asterisks); scale bar: 20 µm. (**B**) A significant decrease in the number of goblet cells counted in serial sections through the conjunctiva (230 sections/lid) and (**C**) the percentage of goblet cells per histological section ((ctrl) = 6, n (OPN) = 15) showing changes in apical vesicles. (**D**) Electron micrographs of a control (ctrl) and 10-day-treated OPN goblet cell with fused apical secretory vesicles (asterisks). The statistic is the mean + SEM; *** *p* < 0.001, **** *p* < 0.0001 (*t*-test).

**Figure 12 ijms-26-01884-f012:**
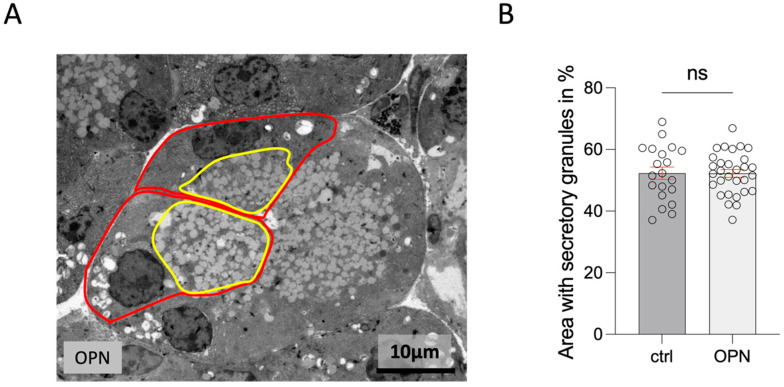
(**A**) Transmission electron micrograph of an endpiece of the lacrimal gland (cells with nuclei in the section marked in red). The apical area containing secretory vesicles is marked in yellow. (**B**) Diagram showing that the percentage of the area with secretory vesicles is not changed in OPN mice compared to the controls. The statistic is the mean + SEM; ns = not significant (*t*-test).

**Figure 13 ijms-26-01884-f013:**
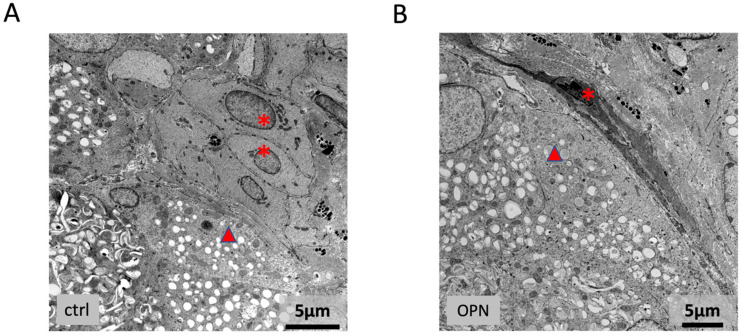
Meibomian gland. (**A**) Control (ctrl) and (**B**) OPN mouse transmission electron micrograph of a smooth muscle cell (asterisks). After 10 days of treatment with oxaliplatin, the muscle cells showed enlarged cisternae of the rER and thickening of the myofilaments. The arrowhead points to a basal meibocyte.

**Figure 14 ijms-26-01884-f014:**
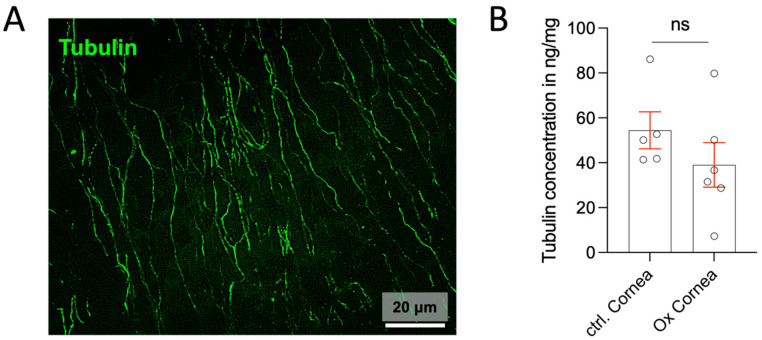
(**A**) Whole mounts of the cornea stained (green) for tubulin. No differences in the staining of subendothelial nerves were seen between the treated and control mice. (**B**) ELISA for tubulin in corneal tissue (n = 6–10). The raw data were relative to the expression of the control; the statistic is the mean + SEM relative to the mean of the control (n = 8); ns = not significant (*t*-test).

**Figure 15 ijms-26-01884-f015:**
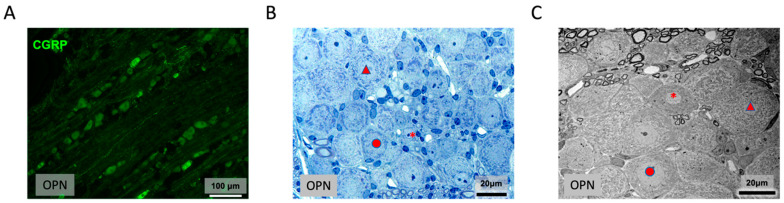
Trigeminal ganglion of the OPN mice (OP): (**A**) Immunohistochemical staining (green) for CGRP. (**B**,**C**) A 1 μm section (**B**) and transmission electron micrograph (**C**) through the trigeminal ganglion of the OPN mice with small (asterisks), medium (circles), and large (arrowheads) neuronal cells. No differences in the morphology of the ganglia were seen between the treated and control mice.

**Table 1 ijms-26-01884-t001:** Primary antibodies used for immunofluorescence.

Antibody	Manufacturer	ID	Kryo(K)/ Wholemount (W)	Dilution
Rabbit anti-alpha smooth muscle actin antibody	Abcam, Cambridge, UK	EPR5368	K	1:1000
Rabbit anti-ATP13A2/PARK9	Bioss, Woburn, MA, USA	bs-11708R	K	1:50
Rabbit anit-cytokeratin 1	EnoGene, New York, NY, USA	E301319	K	1:100
Rabbit anti FAM44A/Bod1l	ThermoFisher Scientific, Rockford, IL, USA	PA559504	K	1:100
Rabbit anti-lactoferrin	Invitrogen, ThermoFisher Scientific, Rockford, IL, USA	PA5-95513	K	1:300
Rabbit anti lactoperoxidase (LPO)	Abcam, Cambridge, UK	PAA296Mu01	K	1:100
Rabbit anti-CGRP	Biotrend, Köln, Germany	ZO5177	W	1:500
Mouse antib-3-tubulin	R&D, Minneapolis, MN, USA	MAB1196	W	1:100

**Table 2 ijms-26-01884-t002:** Oligonucleotide real-time PCR.

Gene	Name	Sequence (5′ → 3′)	Product (bp)
*Atp13a2*	ms ATP13A2_real_fw	GGC CCT CTA CAG CCT GAC T	65
	ms ATP13A2_real_rev	CCA GGT TGG TGT TGA TTG TG	
*Bod1l*	ms BOD1L_real_fw	GAG CCA CAA GAT CAG C	72
	ms BOD1L_real_rev	TTG CAC GTG TAG ATG GCT TG	
*Krt1*	ms KRT1_real_fw	TTT GCC TCC TTC ATC GAC A	108
	ms KRT1_real_rev	GTT TTG GGT CCG GGT TGT	
*LPO*	ms LPO_real_fw	CCC CTG CAC ACT GTC TTT TT	75
	ms LPO_real_rev	CCC CTG CAC ACT GTC TTT TT	
*Ltf*	ms LTF_real_fw	TGG GAA AGG AGT ACG TCA TAG C	82
	ms LTF_real_rev	GTA AGA AAA GCG CAG GCT TC	
*18S*	18S human/mouse real time/SYBR_fw	GGT GCA TGG CCG TTC TTA	69
	18S human/mouse real time/SYBR_rev	TGC CAG AGT CTC GTT CGT TA	
*β-Aktin*	β-Aktin_fw	GAT CCT CAC CGA GCG CGG CTA CA	296
	β-Aktin_rev	CGC GAT GTC CAC GTC ACA CTT CA	

**Table 3 ijms-26-01884-t003:** List of used TaqMan^®^ gene expression assays.

Target	Gene	Article Number	Company
*ATF3*	Activating transcription factor 3	Mm00476033_m1	Thermo Fisher
*GAPDH*	Glceraldehyde-3-phosphate dehydrogenase	Mm99999915_g1	Thermo Fisher
*MMP9*	Matrix metallopeptidase 9	Mm00442991_m1	Thermo Fisher
*NOX2*	NADPH oxidase 2	Mm01287743_m1	Thermo Fisher
*NOX4*	NADPH oxidase 4	Mm00479246_m1	Thermo Fisher
*XDH*	Xanthine dehydrogenase	Mm00442110_m1	Thermo Fisher

## Data Availability

Data is contained within the article.

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
