# Peer review of "Ocular Surface Changes Differ Significantly Between Oxaliplatin- and Diabetes-Induced Polyneuropathy"

_ijms, 2025, doi:10.3390/ijms26051884_

Round 1
Reviewer 1 Report
Comments and Suggestions for Authors
The present experimental study is intriguing, and much work has been done. The same authors with the same methods published the article ‘’Ocular surface changes in mice with streptozotocin-induced diabetes and diabetic polyneuropathy’’ (ref 9). However, in that study, there was a diabetic ‘’milieu’’. My question for the present study is: were the animals healthy before the oxaliplatin-induced changes? Or did they suffer from any tumor? The health status of the animals is crucial, as it could significantly impact the results of the present study. Because some tumors secrete many substances that could affect many other systems or organs, it is important to clarify this point.
Moreover, I saw pictures from electron microscopy in both studies. Did the authors find any changes in the basement membrane thickness in nerves?
The authors are commended for their appropriate methodology. The manuscript is well written, and the discussion/conclusions are acceptable.
Overall, the data are of interest and provide valuable insights into the subject.
Author Response
Comments to reviewer 1:
- My question for the present study is: were the animals healthy before the oxaliplatin-induced changes? Or did they suffer from any tumor? The health status of the animals is crucial, as it could significantly impact the results of the present study.
We thank the reviewer for his comments and have expanded the section on animals in the Materials and Methods section in response to his questions. Line 407-416
For the experiments, we used healthy adult C57/Bl6N wild-type mice that were tumor-free before oxaliplatin treatment. They did not receive any treatment other than oxaliplatin, which was administered solely to induce oxaliplatin-induced neuropathy, following the protocol described by Nassini et al. 2011 (1).The health status of the animals was monitored daily following oxaliplatin treatment. No changes in behavior, signs of bleeding, inflammation, or discomfort were observed. Additionally, the animals were weighed daily, and none of the treated mice exhibited a decrease in body weight after oxaliplatin treatment.
References:
Nassini R, Gees M, Harrison S, De Siena G, Materazzi S, Moretto N, et al. Oxaliplatin elicits mechanical and cold allodynia in rodents via TRPA1 receptor stimulation. Pain. 2011;152(7):1621-31.
- Moreover, I saw pictures from electron microscopy in both studies. Did the authors find any changes in the basement membrane thickness in nerves?
At this early stage of the disease, we did not see ultrastructural basement membrane changes of the nerves, the capillaries or the perineuronal sheaths. We have specifically looked for basement membrane changes, but we could not detect them in either the OPN or the DPN model.

Reviewer 2 Report
Comments and Suggestions for Authors
The manuscript presents an intriguing comparison of ocular surface changes associated with oxaliplatin-induced polyneuropathy (OPN) and diabetes-induced polyneuropathy (DPN) in a murine model. The topic is relevant and offers potential clinical implications, particularly for early diagnostics and understanding of the pathophysiology of polyneuropathies. The manuscript is well-structured, but several areas require refinement to enhance clarity, scientific rigor, and linguistic precision.
The arguments are generally well-founded, supported by robust data from morphological, biochemical, and behavioral analyses. The study offers novel insights, especially in identifying differential ocular surface changes and tear film protein alterations in OPN compared to DPN. The methodology is comprehensive, employing diverse techniques such as immunohistochemistry, electron microscopy, and proteomic analysis. Nonetheless, some methodological details are limited. For example, the sample size justification and statistical analysis details need elaboration to ensure reproducibility.
Here’s my suggestions for line:
· Line 45: "it is impossible to predict a patient’s risk outcome" could be improved to "predicting a patient's risk remains challenging."
· Line 359: Clarify "all animal experiments were performed with 8- to 12-week-old C57BL/6NRj male mice." Consider specifying the number of animals used.
· Line 509: Expand on statistical methods, particularly regarding multiple comparison corrections.
· Line 307: Replace "but possibly due to reduced axoplasmic flow with reduced energy." with "potentially due to impaired axoplasmic transport leading to decreased energy availability."
· Line 325: "The reason why the trigeminal ganglia at this early stage of OPN appear normal is not yet known." Suggest rephrasing to "The mechanisms underlying the preserved morphology of trigeminal ganglia in early-stage OPN remain unclear."
· Line 522: Enhance the final sentence to emphasize clinical implications: "These findings can facilitate specific, early diagnosis of PN in clinical practice and guide targeted treatment strategies for DED."
- The % of plagiarism is very high (49%), please revise it.
Comments on the Quality of English LanguageMinor editing is required
Author Response
Comments to reviewer 2:
- Line 45: "it is impossible to predict a patient’s risk outcome" could be improved to "predicting a patient's risk remains challenging."
We thank you for pointing this out and have revised the section. Line 49
- Line 359: Clarify "all animal experiments were performed with 8- to 12-week-old C57BL/6NRj male mice." Consider specifying the number of animals used.
We thank the reviewer for his comments and have expanded the section on animals in the Materials and methods. Eight animals per group (control vs. OPN) were used for this study. Line 415
- Line 509: Expand on statistical methods, particularly regarding multiple comparison corrections.
We have expanded on statiscal methods. Line 566-576
- Line 307: Replace "but possibly due to reduced axoplasmic flow with reduced energy." with "potentially due to impaired axoplasmic transport leading to decreased energy availability."
We have revised the section according to your suggestion: Line 338
- Line 325: "The reason why the trigeminal ganglia at this early stage of OPN appear normal is not yet known." Suggest rephrasing to "The mechanisms underlying the preserved morphology of trigeminal ganglia in early-stage OPN remain unclear."
Thank you, we have changed the text accordingly. Line 357- 359
- Line 522:Enhance the final sentence to emphasize clinical implications: "These findings can facilitate specific, early diagnosis of PN in clinical practice and guide targeted treatment strategies for DED."
We have revised the text accordingly.
- The % of plagiarism is very high (49%), please revise it.
Most of the sentences declared as plagiats are similar to those used in our previous paper describing the changes in the sciatic nerves, spinal and trigeminal ganglia as well as the ocular surface changes in mice with diabetic polyneuropathy: ‘Ocular surface changes in mice with streptozotocin-induced diabetes and diabetic polyneuropathy’. For comparison of the two PN-models, we evaluated the OPN-animals using the same methods described in that paper. We have taken care to cite the sources correctly to ensure transparency and scientific integrity. To take this into account, we have revised the sections marked in green.

Round 2
Reviewer 2 Report
Comments and Suggestions for Authors
My previous comments were well addressed.